# Functional Inhibition of Valosin-Containing Protein Induces Cardiac Dilation and Dysfunction in a New Dominant-Negative Transgenic Mouse Model

**DOI:** 10.3390/cells10112891

**Published:** 2021-10-26

**Authors:** Xiaonan Sun, Ning Zhou, Ben Ma, Wenqian Wu, Shaunrick Stoll, Lo Lai, Gangjian Qin, Hongyu Qiu

**Affiliations:** 1Center for Molecular and Translational Medicine, Institute of Biomedical Science, Georgia State University, Atlanta, GA 30303, USA; xsun13@gsu.edu (X.S.); myuanchao@163.com (B.M.); wwenqian@gsu.edu (W.W.); lucylai1983@gmail.com (L.L.);; 2Division of Physiology, Department of Basic Sciences, School of Medicine, Loma Linda University, Loma Linda, CA 92318, USA; zhouning@tjh.tjmu.edu.cn (N.Z.); sstoll@students.llu.edu (S.S.); 3Department of Biomedical Engineering, School of Medicine and School of Engineering, University of Alabama at Birmingham, Birmingham, AL 35294, USA; gqin@uab.edu

**Keywords:** valosin-containing protein, dilated cardiomyopathy, nuclear translocation, co-factor interaction, P47, nuclear protein localization protein 4, ubiquitin fusion degradation protein 1

## Abstract

Valosin-containing protein (VCP) was found to play a vital protective role against cardiac stresses. Genetic mutations of VCP are associated with human dilated cardiomyopathy. However, the essential role of VCP in the heart during the physiological condition remains unknown since the VCP knockout in mice is embryonically lethal. We generated a cardiac-specific dominant-negative VCP transgenic (DN-VCP TG) mouse to determine the effects of impaired VCP activity on the heart. Using echocardiography, we showed that cardiac-specific overexpression of DN-VCP induced a remarkable cardiac dilation and progressively declined cardiac function during the aging transition. Mechanistically, DN-VCP did not affect the endogenous VCP (EN-VCP) expression but significantly reduced cardiac ATPase activity in the DN-VCP TG mouse hearts, indicating a functional inhibition. DN-VCP significantly impaired the aging-related cytoplasmic/nuclear shuffling of EN-VCP and its co-factors in the heart tissues and interrupted the balance of the VCP-cofactors interaction between the activating co-factors, ubiquitin fusion degradation protein 1 (UFD-1)/nuclear protein localization protein 4 (NPL-4) complex, and its inhibiting co-factor P47, leading to the binding preference with the inhibitory co-factor, resulting in functional repression of VCP. This DN-VCP TG mouse provides a unique functional-inactivation model for investigating VCP in the heart in physiological and pathological conditions.

## 1. Introduction

Valosin-containing protein (VCP), also known as p97 in mammals and CDC48 in S. cerevisiae, is a member of the type II AAA (ATPases Associated with various cellular Activities) family, which is ubiquitously expressed in cells [1,2,3]. It has been shown that VCP regulates multiple cellular functions, including proteasome degradation, membrane fusion, mitochondrial function, and autophagy in various tissues, and have been found to be involved in various diseases such as cancer, bone disease, and neurodegenerative diseases [3,4,5,6,7]. Specific genetic mutations of VCP have been associated with a multisystem degenerative disorder termed inclusion body myopathy, which is associated with Paget’s disease of bone and frontotemporal dementia (IBMPFD), encompassing muscle, brain, and bone, and with other diseases such as Parkinsonism and dilated cardiomyopathy (DCM) [8,9,10,11,12]. The mechanism by which these mutations cause diseases remains controversial [13,14].

Our previous studies have demonstrated that VCP was decreased in the hypertensive rat hearts and acts as a driving force for the cardiac pathogenesis caused by pressure overload [15]. Reciprocally, cardiac-specific overexpressing VCP protects mouse hearts against pressure-overload-induced pathological hypertrophy [15,16,17] and heart failure [18], and it also protects against stress-induced cardiomyocyte death in vitro and reduced ischemic injury in vivo [19,20,21]. These results together indicated that VCP plays a critical protective role in the heart under stress conditions. However, its essential functions in the heart at the physiological state remain largely unknown. Since the VCP knockout in mice is embryonically lethal [22], functional inactivation of VCP would provide a powerful approach to determine whether VCP is necessarily required for cardiac function in the heart.

It has been known that VCP’s function mainly depends on its binding with its co-factors. Despite the complexity of the protein interactions of VCP in various cells, a few co-factors have been characterized and found to form functional complexes with VCP in some tissues, particularly the ubiquitin fusion degradation protein 1 (UFD-1), nuclear protein localization protein 4 (NPL-4), and p47. VCP appears to perform different cellular functions by binding to its associated co-factors [23]. For example, it has been postulated that UFD-1 and NPL-4 act as heterodimers to form a functional complex by binding to VCP, involved in the retro-translocation of misfolded proteins and VCP nuclear translocation [23,24,25,26,27]. On the other hand, P47 forms a tight, stoichiometric complex with cytosolic VCP, participating in inhibitory functions in VCP activity [28,29]. However, the interaction of VCP with these co-factors and their effects on the heart are unclear.

Structurally, VCP consists of 806 amino acids and has a molecular weight of 97 kDa, and contains four main domains, including an N-domain followed by two consecutive ATPase domains (D1 and D2) and an unstructured C-terminal tail (CT). It has been shown that D1 and D2 domains are responsible for maintaining the hexamer structure and generating energy by ATP hydrolysis, respectively, while the N-domain is involved in the substrate’s recognition of, and interaction with, other co-factors [30,31,32]. N-domain consists of 187 amino acids followed by a 21 amino acids short N-D1 linker. It has been shown that VCP-mediated ATPase activity requires both N-domain and the N-D1 linker. The conformation of the N-domain and linker relative to the D1-D2 hexamer is directly correlated with p97 ATPase activity [33]; for example, the binding of p47 to VCP via the N-domain causes inhibition of p97 ATPase activity [34]. N-domain is also considered to participate in the regulation of VCP nuclear entry, while the CT may involve VCP nuclear export regulation [35]. Interestingly, it has been shown that most of the reported IBMPFD-causing mutations occur in the N-domain, suggesting an essential role of this domain in VCP function [36]. Our previous in vitro study showed that overexpressing a mutated VCP, in which the N-domain has been deleted, eliminates the protective ability of VCP in neonatal cardiomyocytes [21]. This finding indicated that VCP N-domain plays a crucial role in cardiac protection. However, the role of VCP N-domain in the heart in vivo and the regulatory mechanisms were not explored. 

Based on our previous results from the in vitro studies in cardiomyocytes and the reports from other tissues/cells, we hypothesized that VCP N-domain is necessary for the VCP function in the heart, and the deletion of this N-domain acts as a dominant-negative mutation of VCP (DN-VCP), inducing a functional inhibition of the endogenous VCP (EN-VCP), leading to adverse effects on the heart. To test this hypothesis, we generated a cardiac-specific transgenic mouse (TG) overexpressing a mutant of VCP bearing a deletion of N-domain of VCP to test whether overexpressed DN-VCP exhibits a functional inhibition on EN-VCP. We also determined whether VCP is necessary for cardiac growth and function during the physiological condition. By investigating the temporal effects of DN-VCP on cardiac morphology and contractile function in the TG mouse at baseline, we found that DN-VCP impaired VCP activity, resulting in a cardiac deterioration during the aging transition. We also explored the molecular mechanisms underlying the adverse effects of DN-VCP observed in mouse hearts. 

## 2. Materials and Methods

### 2.1. Animal Models

A cardiac-specific DN-VCP TG mouse was generated with an FVB background as described previously [15]. This DN-VCP mutant lacks the first 600 nucleotides of the coding human VCP sequence [21]. A construct harboring the coding sequence of DN-VCP was ligated downstream of the cardiac-specific promoter of an α-myosin heavy chain (αMHC). The mouse colony was maintained by mating TG mice with FVB wild-type (WT) mice. Studies were performed at two age points, i.e., 10–12 weeks (young) and 52–54 weeks (older). Litter-matched or age-matched WT mice were used as controls. Male and female mice were included. All animal procedures were performed in accordance with the NIH guidance (*Guide for the Care and Use of Laboratory Animals*, revised 2011) and the protocols approved by the Institutional Animal Care and Use Committee of Loma Linda University and Georgia State University. 

### 2.2. Echocardiography

Cardiac function and structure were determined by echocardiography using a GE Logiq Evet machine with a 13-MHz probe. The M-mode recording was made at the middle level of the left ventricle (LV) in a left parasternal long-axis view. All animals were anesthetized by using 2% isoflurane, and the heart rates of all mice were maintained above 450 bpm while being imaged. LV structure and function were measured as described previously [15,37], including LV anterior wall thickness at end-diastole and end-systole (LVAWd, LVAWs), and LV posterior wall thickness at end-diastole and end-systole (LVPWd, and LVPWs), LV internal dimensions at end-diastole (LVEDd), and end-systole (LVEDs), heart rate (HR), LV ejection fraction (LVEF), and fractional shortening (LVFS). 

### 2.3. Protein Extraction and Subcellular Fraction and Detection

Total proteins and subcellular fractions were extracted as described previously [15,38,39]. In brief, for total protein extraction, heart tissues or transfected cells were homogenized with the homogenizer (OMNI, bead mill homogenizer) in RIPA buffer with EDTA (Boston bioproducts, BP-115D) and protease inhibitor (Roche,11836153001) and phosphatase inhibitor (Roche, 04906837001) [15,38]. Centrifuge at 14,000 rpm at 4 °C for 15 min. 

For subcellular fraction, according to the protocol of the manufacturer (Abcam, Nuclear Extraction kit, ab113474), cells or heart tissues were separated into cytoplasmic protein and nuclear protein [15,38]. In brief, mouse heart tissues or transfected cells were homogenized in the pre-extraction buffer. Centrifuge at 12,000 rpm for 10 min at 4 °C, transfer the supernatant to a new tube. Incubate the nuclear pellet in extraction buffer with vortex, sonicate the extract 3 times, each time 10 s. Centrifuge the suspension at 14,000 rpm for 10 min at 4 °C.

Protein levels were detected by Western blotting, as described previously [15,20,21,38,40], with the LI-COR Odyssey infrared imaging system (LI-COR Biosciences, Lincoln, NE, USA). Briefly speaking, total protein and subcellular fractions were quantified by Pierce BCA protein assay (Thermo Fisher Scientific, Waltham, MA, USA, 1859078), loaded onto SDS-PAGE gels, and transferred to nitrocellulose membranes. Primary antibodies (anti-VCP-C terminal (Invitrogen Thermo Fisher Scientific, Waltham, MA, USA. MA3–004), anti-P47 (Santa Cruz Biotechnology, Dallas, TX, USA, sc-365215), anti-UFD-1(Cell Signaling Technology, Danvers, MA, USA, 13789 s), anti-NPL-4 (Cell Signaling Technology, Danvers, MA, USA, 13489 s), anti-GAPDH (Cell Signaling Technology, Danvers, MA, USA; 2118), anti-Histon 3 (Cell Signaling Technology, Danvers, MA, USA. 4499) antibodies), and the corresponding secondary antibody (LI-COR, Lincoln, NE, USA, IRDye800CW, IRDye680LT) were probed for the target proteins. The image was analyzed by chemiluminescence (LI-COR, Lincoln, NE, USA, odyssey CLx).

### 2.4. Immonuoparticipation (IP) and Subsequent Western Blotting

IP and Western blotting were performed as described previously [15,20,21,38,40]. For immunoprecipitation, protein G beads were prepared and incubated with the specific antibodies, including anti-Flag (Sigma, F1804), anti-VCP-C terminal (Invitrogen, MA3-004), anti-P47(Santa Cruz, sc-365215) for 1 h, and the extracts were incubated with protein G beads at 4°C overnight. Interacting proteins were then detected by Western blotting. 

After incubation with the specific primary antibodies (anti-Flag (Sigma-Aldrich, St. Louis, MO, USA. F1804), and other antibodies mentioned above, including anti-VCP-C terminal, anti-P47, anti-UFD-1, anti-NPL-4; and the corresponding secondary antibodies; the detection was performed as descripted above.

### 2.5. ATPase Activity

Heart tissue samples were homogenized on ice with 200 µl assay buffer, and ATPase activities were measured using the ATPase/GTPase Activity Assay Kit (Sigma–Aldrich, MAK-113) according to the manufacturer’s protocol. In brief, tissue samples were homogenized on ice with assay buffer. Centrifuge at 14,000 rpm for 10 min to remove insoluble material; then, the reaction mix (assay buffer and ATP) was added to each sample, background blank, and negative controls. The phosphate standards for colorimetric detection and each sample were run in duplicate. After 30 min of incubation at room temperature, the stopping buffer was added to terminate the enzyme reaction and generate a colorimetric product. The enzyme activity resulted from the reading of absorbance at 600–660 nm and was calculated by the standard curve. 

### 2.6. Cell Culture and Transfection

HEK293T cell line was cultured in DMEM (Corning, 10-013-CV), with 5% fetal bovine serum (Corning, 35011CV) and 1% streptomycin/penicillin (Gibco™ 15070063). Cells were kept in the cell incubator at 37 °C with 5% CO^2^. Two plasmids of PcDNA3.1 vector harboring full-length VCP (FL-VCP) and DN-VCP were generated with a specific tag Flag at the C-terminal to overexpress FL-VCP and DN-VCP in 293 cells; a plasmid with PcDNA3.1 vector-only was used as a control. FL-VCP-Flag and DN-VCP-Flag plasmids were transfected in 6-well plates with transfection reagent based on the manufacturer’s protocol. The protein expression was determined by Western blot analysis.

### 2.7. Immunofluorescent Staining

Immunofluorescent staining for VCP was performed in HEK293T cells for measuring the subcellular distribution of EN-VCP or DN-VCP inside the cells. Briefly, HEK293T cells were transfected with PcDNA 3.1 vector plasmids harboring FL-VCP-Flag or DN-VCP-Flag, for 48 h. The PcDNA3.1 empty vector plasmid was used as a control. The cells were then fixed with 3.7% formaldehyde in PBS, permeabilized in 0.1% Triton X-100 in PBS. After fixation and permeabilization, cells were washed three times with PBST (PBST: 0.05% tween-20/PBS) containing 1% bovine serum albumin (BSA) and incubated in the specific first antibody against Flag (Sigma, F1804) at a dilution of 1:100 overnight at 4 °C. Alexa Fluor-conjugated secondary antibody (1:200) was used for detection under standard immunofluorescence staining protocol. The images of the staining were taken from the transmitted cells under microscopy (Olympus, IX73).

### 2.8. Statistical Analysis

Data are presented as means + standard error (SEM) for the number per group indicated in each figure legend. Comparisons between the two groups were performed using an unpaired Student’s *t*-test. A value of *p* < 0.05 was considered significant.

## 3. Results

### 3.1. DN-VCP Does Not Alter EN-VCP Expression but Induces a Cardiac Dilation in Young TG Mice

To determine the role of VCP in the heart, we first identified and characterized the newly generated cardiac-specific DN-VCP TG mouse. As illustrated in Figure 1a, we used the same construct as we tested previously in the in vitro study [21], in which a mutant VCP bearing a deletion of N-domain was overexpressed under the a-MHC promoter. The TG mouse genotyping was identified by PCR (Figure 1b). The expression of DN-VCP protein in the heart was detected by Western blotting at a molecular weight (MW) of 75 kDa, while the EN-VCP was observed at 97 kDa (Figure 1c). The relative level of the DN-VCP to EN-VCP is about 62% in TG mice at the age of 10 weeks (Figure 1d), and there is no gender difference between males and females (Figure 1d). The overexpression of DN-VCP in the hearts does not significantly affect the EN-VCP expression in the TG mice compared to WT mice at this age (Figure 1d).

The cardiac morphology and function were further determined in these mice by echocardiography at the age of 10 weeks. As shown in Figure 1e, there is no significant difference in LV wall thickness between WT and TG mice in terms of LVAWd, LVAWs, LVPWd, and LVPWs. However, LV internal dimensions (LVEDd and LVEDs) were increased in DN-VCP TG mice compared to their age-matched WT mice (*p* < 0.01) (Figure 1f), indicating a cardiac dilatation in the TG mice at this early age stage. The cardiac function is comparable between two groups at this age, represented by a similar heart rate (Figure 1g), LV EF, and FS (Figure 1h). In addition, the overexpression of DN-VCP in the heart does not alter mouse mortality or the gender distribution of their offspring. There is no significant difference in the heart’s development in DN-VCP TG compared to their litter-matched WT mice in terms of the cardiac anatomic structure, shape, and size (Figure 1i) at 10 weeks.

### 3.2. DN-VCP Induces a Progressive Dilated Cardiomyopathy and Declined Cardiac Function during the Aging Transition

To investigate the potential chronic effect of DN-VCP in the heart, we maintained both DN-VCP TG and WT mice up to 52–54 weeks and measured their cardiac morphology and function by echocardiography. As shown in Figure 2a–c, compared to the age-matched WT, DN-VCP TG mice at this older age exhibited a significantly lower LV wall thickness (Figure 2a) and a larger LV internal dimension (Figure 2b), indicating a remarkable dilated cardiomyopathy in DN-VCP TG mice at this older age. In addition, DN-VCP TG mice exhibited a declined LV, EF, and FS compared to their age-matched WT mice (Figure 2c), indicating a decreased cardiac function. Furthermore, although the body weights were comparable between the two groups, the heart weight in DN-VCP TG mice was significantly greater than the WT mice, as normalized to body weight (Figure 2d) or tibia length (Figure 2e).

Together, these data indicate that the chronic overexpression of DN-VCP in TG mouse hearts induced a progressive cardiac deterioration, resulting in dilated cardiomyopathy and cardiac dysfunction during the aging transition.

### 3.3. DN-VCP Aggravates the Age-Related Alterations of the EN-VCP and Its Co-Factors Expression in the Heart

To determine whether the molecular mechanism involved in the DN-VCP TG caused cardiac deterioration, we first tested the expression of DN-VCP and EN-VCP in the DN-VCP TG hearts at both young and older ages compared to their age-matched WT mice (Figure 3a). As shown in Figure 3b, EN-VCP expression in the heart exhibited an age-related declining trend in the older WT mice compared to the young WT mice, and this age-associated decrease in EN-VCP was more remarkable in DN-VCP TG mice between young and older ages (*p* < 0.01, vs young DN-VCP TG mice) (Figure 3b). However, there was no significant difference in the EN-VCP levels between DN-VCP TG and WT mice at either age group, further indicating that DN-VCP overexpression did not dramatically affect the EN-VCP expression in the heart. We also noticed that the DN-VCP expression was slightly decreased in the older TG mouse hearts compared to the young TG mice (Figure 3c).

We next tested whether DN-VCP affects the expression of VCP-function-associated proteins in the heart (Figure 3d). We first measured the expression of UFD-1 and NPL4, which were considered the functional co-factors to activate the VCP upon their binding. As shown in Figure 3e, UFD-1 was slightly decreased in older WT mouse hearts compared to young WT mice (*p* = 0.06), but this age-related decline in UFD-1 was more significant in old DN-VCP TG mice compared to their corresponding young mice (*p* < 0.001, vs. young DN-VCP TG mice) (Figure 3e). Compared to their age-matched WT mice, the cardiac expression of UFD-1 was notably increased in DN-VCP TG mice at a young age (*p* = 0.02) but not at an older age. There were no significant differences in the NPL4 protein level between DN-VCP TG and WT mice for either age group (Figure 3f).

Reciprocally, P47, another VCP co-factor potentially playing an inhibitory effect of VCP, was substantially increased in an older WT mouse heart compared to young WT mice; this age-related increase in P47 was even more pronounced in old DN-VCP TG mice compared to their corresponding young TG mice (Figure 3g). There is no significant difference in P47 protein levels between DN-VCP TG and WT at a young age; however, P47 was significantly higher in older DN-VCP TG mice compared to their age-matched WT mice.

Together, these data indicate an age-related alteration in EN-VCP and its functionally associated proteins in the WT mice during the aging transition; these changes were aggravated in DN-VCP TG mice. Interestingly, ENVCP and UFD-1 exhibit a similar pattern to the changes between WT and TG mice during the aging transition (Figure 3b,e), while the P47 showed an opposite change to them in these mice (Figure 3g).

### 3.4. DN-VCP Attenuates Age-Related Cytoplasmic/Nuclear Shuffling of EN-VCP and Its Co-Factors in TG Mouse Hearts

We then tested whether DN-VCP affects the subcellular distribution of EN-VCP and its co-factors in the heart during the aging transition. Since both EN-VCP and DN-VCP proteins co-existed in the TG mouse hearts, and the cellular distribution of DN-VCP was unknown, we first performed an in vitro study to determine the cellular distribution of DN-VCP by overexpressing the DN-VCP in HEK293T cell line via a plasmid vector carrying a Flag tag sequence, and the plasmid with the full-length sequencing of WT-VCP (FL-VCP) or the vector only were used as controls. An antibody against Flag was used to identify the exogenous WT-VCP (FL-VCP) and DN-VCP without the influence of the endogenous VCP in the cells.

As shown in Figure 4a, while the overexpressed FL-VCP was distributed in both cytoplasmic and nuclear fractions, overexpressed DN-VCP was detected mainly in the cytoplasm, indicating the failure of the nuclear entry of DN-VCP. Cyto-immunostaining with antibody against Flag further confirmed the loss of the signal of DN-VCP in the nucleus of the cells, while the FL-VCP signals were detected in both nucleus and cytoplasm (Figure 4b). These data indicated that the deletion of VCP N-domain abolished its nuclear translocation inside the cells.

We further confirmed the observation in these cells in DN-VCP TG mouse hearts. The nuclear and cytoplasmic fractions were isolated from mouse heart tissues, and the corresponding protein levels were detected by Western blotting separately. As shown in Figure 4c, while EN-VCP was detected in both nuclear and cytoplasmic fractions, overexpressed DN-VCP was predominantly detected in the cytoplasm, with a negligible amount in the nuclear fraction. This result confirmed the essential role of the VCP N-domain in nuclear translocation in the heart tissues.

Compared to the young WT mice, the EN-VCP level was significantly decreased in the cytoplasmic fraction but increased in the nuclear fraction in older WT mice (Figure 4d), indicating an aging-associated increase in VCP cytoplasm to nuclear shuffling, which might be a response to the decrease of total EN-VCP in the hearts of older WT mice. However, this shuffling was attenuated in DN-VCP TG mice, leading to a significant reduction in EN-VCP in the nuclear fraction of DN-VCP TG older mice compared to their age-matched WT mice (Figure 4d). These data indicate an extra repressive role of DN-VCP on the aging-associated adaption in VCP cytoplasmic/nuclear shuffling in older TG mice, but not in young DN-VCP mice (Figure 4d).

In addition, we tested whether DN-VCP affects the cytoplasmic/nuclear shuttling of VCP co-factors in the heart. As shown in Figure 4e–g, UFD-1 and P47 exhibited a decrease in cytoplasmic fraction but an increase in the nuclear fraction of the heart tissues in the older WT compared to the young WT mice, indicating an enhanced cytoplasm to nuclear shuffling during aging, consisting with the age-related alteration in EN-VCP in WT mice. NPL-4 was significantly decreased in the cytoplasmic fraction in older WT mice vs. young WT mice, but no significant change was found in the nuclear fraction during the aging transition in WT mice. However, these aging-related cytoplasmic/nuclear shuttling in VCP-cofactors have also attenuated in the DN-VCP TG older mice (Figure 4e,f).

These data together indicate that VCP N-domain is necessary for VCP nuclear translocation in the heart. Overexpression of DN-VCP impaired cytoplasm to the nuclear shuttling of EN-VCP and its associated proteins in the heart, particularly in older mice.

### 3.5. DN-VCP Disrupts the VCP-Cofactor Interaction and Leads to a Binding Preference of EN-VCP with the Repressive Protein P47

We next determined the influence of DN-VCP on the interaction between EN-VCP and its co-factors in the mouse heart tissues. We first performed additional in vitro experiments to distinguish the VCP-cofactors interaction between EN-VCP and DN-VCP proteins. Immunoprecipitation (IP) was performed using an antibody targeting the Flag to pull down the interacting proteins DN-VCP, respectively so that the cofactor-interaction of DN-VCP could be detected independent of the EN-VCP. The results showed that NPL-4, UFD-1, and P47 interacted with FL-VCP but not with DN-VCP (Figure 5a). This result suggested that the VCP N-domain is essential for the protein interaction of VCP with these co-factors.

We then confirmed this result from the cells by using DN-VCP TG mouse heart tissues. An antibody targeting at VCP-C-terminal was used to pull down the interacting proteins in the cytoplasmic fraction. Interacting proteins were further validated in the heart tissues by Western blotting. We found that VCP interacted with NPL-4, UFD-1, and p47 in the heart tissues of all mouse groups. It is remarkable that the interaction between VCP and P47 was increased in DN-VCP TG mice compared to their age-matched WT mice (Figure 5b). In addition, we used an antibody against P47 to pull down the interacting proteins and found that P47 only interacted with VCP but neither NPL-4 nor UFD-1. Importantly, we noticed that, as both EN-VCP and DN-VCP were detected in the input protein, only EN-VCP, but not DN-VCP, was detected in the IP protein with antibody against p47, indicating that P47 only interacted with EN-VCP but not DN-VCP (Figure 5c). These data from the mouse heart tissues consisted of the observation from the cells. These data together confirmed that DN-VCP lost the interaction with the VCP co-factors and also alteration of the VCP-cofactor interaction, with a preferred interaction with the repressive binding protein p47.

We next determined whether these alterations caused by DN-VCP induced a functional inhibition of EN-VCP by measuring the ATPase activity. We found that ATPase activity was significantly decreased in DN-VCP mice than in their age-matched WT mice in both age points, and this inhibition was more remarkable in older mice (Figure 5d). These data indicated that DN-VCP exhibits a functional inhibition of EN-VCP in the TG mouse hearts.

## 4. Discussion

Our previous studies demonstrated a strong link between the down-regulation of VCP expression and cardiomyocyte hypertrophy and heart failure under pressure-overload stress [15,18]. We also showed that overexpression of VCP protects the heart against cardiac injuries and prevents cardiac deterioration under various stress conditions [15,16,18,19,20,21]. These results indicate that VCP is a crucial protector in the stressed heart. However, despite the high correlation of VCP in various diseases, the role of VCP in the heart under physiological conditions is unknown due to the lack of pertaining models. The present study provides additional evidence indicating that VCP is an essential regulator of cardiac function in physiological conditions, which has been uncharacterized previously. With a newly generated cardiac-specific DN-VCP TG mouse model, we showed that the overexpression of DN-VCP did not affect the expression of EN-VCP in the heart but significantly reduced VCP’s ATPase activity, leading to the functional inhibition of VCP in the heart. In addition, we found that the functional inhibition of EN-VCP induced progressive dilated cardiomyopathy and cardiac dysfunction during the aging transition. These results revealed that DN-VCP yielded an adverse effect on cardiac growth and function via inhibiting EN-VCP activity, offering a dominant-negative mutated model of VCP in the heart [41]. Since the VCP KO mice are lathy [22], this model provides an alternative approach for studying the essential role of VCP in both physiological and pathological cardiac conditions through partial functional inactivation. In addition, our results also revealed the associated mechanisms in the development of dilated cardiomyopathy during the aging transition in DN-VCP TG mice, involving the alterations of VCP’s ATPase activity, co-factor interactions, and cytoplasmic/nuclear shuffling. These results brought new insights into understanding molecular mechanisms in the early stage of cardiomyopathy during the aging transition. They would be beneficial for investigating the strategies preventing aging-related heart diseases.

Although most human dilated cardiomyopathy is idiopathic, ventricle enlargement is one of the typical clinical manifestations, which can either be a primary process of cardiomyopathy or secondary to heart failure. Our results showed that, compared to their age-matched WT mice, overexpression of DN-VCP induced a geometric enlargement of the ventricles at an early age, with a preserved cardiac function but developed cardiac dysfunction with a worsening dilated cardiomyopathy at an older age. These results indicated that the cardiomyopathy induced by the functional inhibition of VCP is a progressive deterioration process, in which the ventricle enlargement acts as an early alteration and initial myocardial remodeling, and the continued ventricle enlargement further elevates both end-systolic and end-diastolic volumes and increases the cardiac loading, leading to the reduction of the cardiac function at an older age. Since the cardiac-specific DN-VCP TG mouse exhibits a phenotype of chronic dilated cardiomyopathy similar to that observed in IBMPFD and other idiopathic DCM, this mouse model provides a unique tool to investigate the molecular mechanisms of cardiomyopathy involved in these complex human disorders, holding great promise for exploring therapeutic targets and thus offering a significant clinical relevance.

Being an AAA-ATPase, VCP is implicated in multiple cellular processes by using the energy from ATP hydrolysis. A functional deficiency of VCP was linked to the pathology of neurons and considered to contribute to impaired DNA repair in multiple polyglutamine diseases [42]. Our results showed that, compared with their age-matched WT, DN-VCP TG mouse hearts exhibited a reduced ATPase activity starting at an early age and worsening at an older age. This progressive alteration is consistent with the development of cardiac geometric enlargement and dysfunction, indicating an association between the reduction of VCP activity and the development of DCM. However, the regulation of ATPase activity of VCP in the heart remains poorly understood. One hypothesis is that VCP basal ATPase activity depends on its interdomain communication or conformation, regulating its ATP binding or hydrolysis. Our results pose the possibility that DN-VCP impaired its interdomain communication due to the deletion of N-domain, and, thus, the loss of the capability of ATP hydrolysis; however, ATP can still bind to DN-VCP’s D1 or D2 domain, which competitively interfere with the binding of ATP to EN-VCP, thus reducing the basal VCP ATPase activity in the DN-VCP TG mouse hearts.

In addition, it has been known that VCP’s activity largely depends on its binding with the specific co-factors, mainly through its N-domain, engaging the ATPase in particular functions. Given the complexity of the protein interactions of VCP in other cells, we focused on two best-characterized co-factors, UFD-1/NPL-4 heterodimer and P47, which have been found to form complexes with VCP in other tissues. A recent study showed that P47, but not the UFD1/NPL4, is involved in VCP-mediated dendritic spine formation and plays a critical role in dendritic spinogenesis [43], indicating an idiosyncratic role of the individual co-factor interaction of the VCP in cellular functions. Furthermore, it has been shown that UFD-1/NPL-4 competes with P47 for binding to VCP in other cells [23,44]. Thus, the binding preference of VCP between P47 and UFD-1/NPL-4 would direct into different cellular pathways, influencing the function of VCP in cells and targeting VCP to various protein machinery. Previous studies have indicated that ATP status influenced the competitive binding in a concentration-dependent manner, regulating the binding of VCP to its different co-factors [45]. Namely, under ATP-limiting conditions, VCP favors binding to P47 over UFD-1/NPL-4, whereas under ATP-abundance, p97/VCP binds preferably to UFD-1/NPL-4 [45]. Our results showed that DN-VCP decreases the total ATPase activity in the mouse heart tissues, explaining the increased interaction of VCP and P47 in the DN-VCP mouse hearts. Interestingly, previous studies have also shown that binding to P47 suppresses the ATPase activity of VCP by restraining the conformational change of the N domain [23,25]. These results indicate a potentially vicious circle in the VCP–P47 binding and the inhibition of the VCP ATP activity, suggesting that binding competition between P47 and UFD-1/NPL-4 to VCP can become heavily biased in specific cells or tissues under pathological conditions [34,43,45]. These studies highlighted the importance of VCP–P47 binding and brought a potential mechanism of progressive cardiac deterioration in the DN-VCP TG mice during the aging transition. Since ATPase activity of VCP was reduced in DN-VCP TG mice at a young age, VCP is more likely to remain bound to P47 when ATP is limiting and thus inhibiting hydrolysis; VCP–P47 binding, in turn, inhibits the ATPase activity of VCP, leading to a worsening VCP function, resulting in cardiac dysfunction in older DN-VCP mice.

Moreover, we also found that UFD-1 was slightly increased in young DN-VCP TG mice than their age-matched WT mice, while the P47 remained unchanged. This increase in the UFD-1 is likely an early compensatory response to the decreased ATPase activity at the early stage. However, in the older mice, UFD-1 was decreased, but the P47 was increased in DN-VCP TG mice compared to their young mice, indicating a decompensation of this mechanism in the older DN-VCP TG mice. These opposite changes in UFD-1 and P47 further aggravated the unbalance of the VCP binding with P47 and UFD-1/NPL-4, which may also contribute to the development of the cardia dysfunction in older DN-VCP TG mice.

Nucleocytoplasmic shuttling is essential for VCP to function spatially in the nucleus and cytoplasm. Although the exact nuclear role of VCP remains largely unknown, some nuclear functions of VCP have been suspected in other cells and species, such as transcriptional control, nuclear envelop formation, cell cycle progression, nuclear formation, and the maintenance of DNA integration [46]. The nuclear role of VCP may also involve regulating the abundance of its partner proteins, modulating the availability of these proteins to participate in metabolic processes in response to DNA damage [46]. Our results revealed that the traffic of VCP between the nucleus and cytoplasm in the hearts underwent a regulated process during the aging transition at physiological conditions. Compared to the young WT mice, the total EN-VCP was slightly decreased, but its nuclear distribution was dramatically increased, and cytoplasmic distribution was decreased in the older WT mice. These age-related changes in the subcellular distribution were also observed in VCP-cofactors, UFD-1, NPL-4, and P47. Our results posit a possibility that the changes in nuclear/cytoplasmic shuffling of these proteins are an adaptive mechanism in response to the increased DNA damage during aging and compensation to the decreased total VCP protein in older WT mice. However, this adaptation was attenuated by the DN-VCP, resulting in a disarranged distribution of EN-VCP and its binding partners. The impairment in the nuclear translocation of VCP and its co-factors may also contribute to cardiac deterioration in older TG mice, declining cardiac function.

As illustrated in Figure 6, several potential mechanisms may explain the impaired VCP nuclear translocation in older DN-VCP TG mice. First, it has been shown that UFD-1/NPL-4 participated in the VCP nuclear translocation by binding to VCP N-domain. Our results showed that DN-VCP loses the capability of the interaction with UFD-1/NPL-4, blocking the DN-VCP nuclear entry process in the heart. These results were confirmed by our in vitro experiments in cells and are consistent with the previous cell-basis studies either on the study of its homolog, Cdc48 in Saccharomyces cerevisiae [47], or using a VCP–GFP chimeric protein in other cells [35]. Secondly, UFD-1 was significantly decreased in older DN-VCP TG mice, which impaired the binding between VCP and UFD-1/NPL-4 and released more free VCP to bind P47. Meanwhile, P47 was increased in the DN-VCP TG old mouse hearts, which increased binding of VCP with P47 in old DN-VCP mice and competitively inhibited VCP’s binding with NPL-4/UFD-1, further attenuating the VCP nuclear translocation. Thirdly, previous studies indicated that ATPase activity is required to regulate the VCP N-domain conformational change during the nuclear entry process. In DN-VCP TG mice, the ATPase activity was decreased, which further attenuates the EN-VCP entry to the nuclear. These together implied that both the unbalanced binding of the VCP with its co-factors and the decrease of ATPase activity contributes to the impaired nucleocytoplasmic shuttling of VCP. On the other hand, it has been reported that the C-terminal region regulates the nuclear export of VCP [35], although the mechanism is unclear yet. Since the overexpressed DN-VCP increases the VCP-CT region in DN-VCP TG mice, this may also participate in the impaired nuclear/cytoplasmic shuffling.

In summary, our results have demonstrated that VCP function plays an essential role in maintaining cardiac function in physiological conditions. DN-VCP confers a functional inhibition of VCP in the heart through competitively interfering EN-VCP’s ATPase activity and unbalancing the VCP’s co-factors interaction and nuclear translocation, leading to cardiomyopathy and dysfunction during the aging transition. Our study provides a unique mouse model to investigate the function of VCP under the physiological state; it is also beneficial for the study of VCP’s protective roles under cardiac stress. The result also provided novel insights into cofactor-specific partners and the diversity of VCP complexes in regulating heart function. With this unique mouse model, our study provides novel evidence for the role of VCP function in the heart at a non-stressed state and the potential mechanisms associated with the development of cardiomyopathy and cardiac dysfunction induced by impaired VCP function. It also brought new insights with which to understand the pathogenesis of complex cardiomyopathy in humans.

## Figures and Tables

**Figure 1 cells-10-02891-f001:**
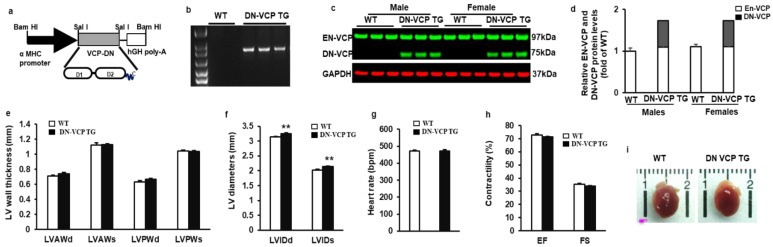
Identification and characterization of cardiac-specific VCP dominant-negative transgenic mouse (DN-VCP TG) at the age of 10 weeks. (**a**). The construct of the cardiac-specific DN-VCP TG mouse model; (**b**,**c**). Identification of DN-VCP TG by genotyping with PCR (**b**) and protein expression in the hearts by Western blots (**c**); (**d**). the relative value of endogenous VCP (EN-VCP) and overexpressed DN-VCP in TG and WT mice of both genders; (**e**–**h**). The morphology and function of DN-VCP TG vs. WT by echocardiography at a young age, in terms of left ventricle (LV) wall thickness including LV anterior and posterior wall thickness at end-diastole and end-systole (LVAWd, LVAWs, LVPWd, and LVPWs) (**e**), LV internal end-diastolic and end-systolic dimensions (LVEDd and LVEDs) (**f**), heart rates (**g**), and contractility reflected by LV ejection fraction (EF) and fractional shortening (FS) (**h**). **, *p* < 0.01 vs. age-matched WT mice. N = 11-12/group; (**i**). The examples of the heart shape and size of DN-VCPTG vs. WT mice at the age of 10 weeks.

**Figure 2 cells-10-02891-f002:**
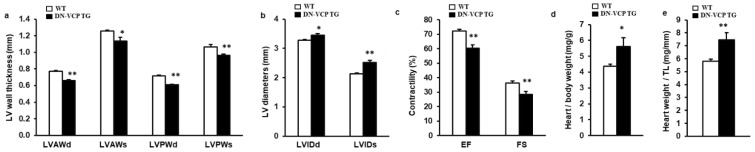
DN-VCP induces a progressive dilated cardiomyopathy and declined cardiac function in the older mice. (**a**–**c**). Cardiac morphology and function in both VCP dominant-negative transgenic (DN-VCP TG) and wild-type (WT) mice at the age of 52 weeks by echocardiography, representing a decreased LV wall thickness, increased LV diameter and declined contractility in DN-VCP TG mice vs. WT. *n* = 8/group; (**d**,**e**). The heart weights normalized to body weight (**d**) and tibia length (TL) (**e**) in DN-VCP TG vs. WT mice. *n* = 4–6/group. *, *p* < 0.05 **, *p* < 0.01 vs age-matched WT mice. LVAWd and LVAWs: LV anterior wall thickness at end-diastole and end-systole; LVPWd and LVPWs: LV posterior wall thickness at end-diastole and end-systole; LVEDd and LVEDs: LV internal dimensions at end-diastole and end-systole; EF and FS: LV ejection fraction and fractional shortening.

**Figure 3 cells-10-02891-f003:**
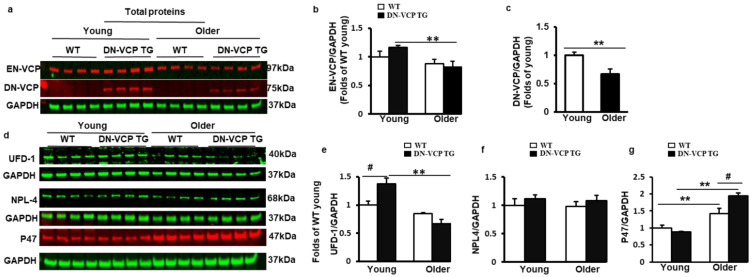
DN-VCP aggravates the age-related alterations in EN-VCP and its co-factors in TG mouse hearts. (**a**). The representative images of Western blots reflect the total protein levels of endogenous VCP (EN-VCP) and dominant-negative VCP (DN-VCP) in DN-VCP TG mice compared to their age-matched WT mice at both young and older ages. GAPDH was used as a loading control; (**b**,**c**). The quantitated values of the total proteins of EN-VCP (**b**) and DN-VCP (**c**) normalized to the GAPDH in the hearts; (**d**). The representative Western blots reflect the total protein levels of VCP’s co-factors in DN-VCP TG mice compared to their age-matched WT mice. GAPDH was used as a loading control of the total proteins; (**e**–**g**). The relative total protein value of the VCP-cofactors, ubiquitin fusion degradation protein 1 (UFD-1) (**e**), nuclear protein localization protein 4 (NPL-4) (**f**), and P47 (**g**), in TG and WT mice. N = 4/group. **, *p* ≤ 0.01 vs. corresponding young mice. #, *p*≤ 0.05, vs. age-matched WT mice.

**Figure 4 cells-10-02891-f004:**
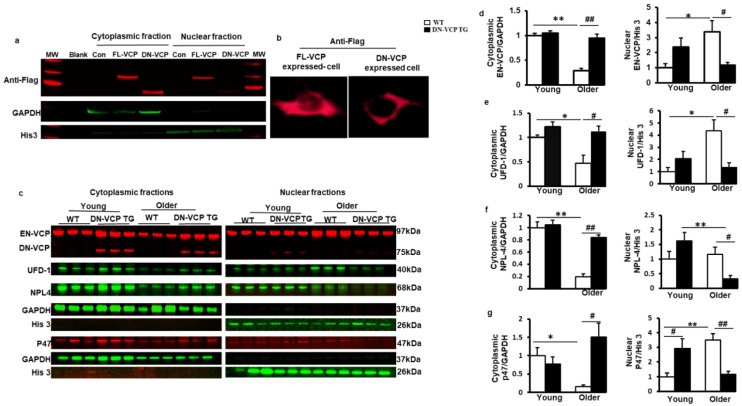
DN-VCP impairs its nuclear localization and attenuates the age-related cytoplasmic/nuclear shuffling of the EN-VCP and its co-factors/adaptor in TG mouse hearts. (**a**,**b**). The subcellular distributions of overexpressed full-length VCP (FL-VCP) and dominant-negative VCP (DN-VCP) in the cytoplasmic and nuclear fractions of the cells, with a specific antibody against the tag-protein-Flag, by Western blots (**a**) and cytoimmunostaining (**b**). Plasmid with vector-only was used as control (Con); (**c**). The representing Western blots reflected the subcellular compartmentalization of the corresponding proteins in the DN-VCP TG mouse heart tissues and compared to their age-matched WT mice at both young and older ages, including endogenous VCP (EN VCP), dominant-negative VCP (DN-VCP) and co-factors, ubiquitin fusion degradation protein 1 (UFD-1), nuclear protein localization protein 4 (NPL-4), and P47. GAPDH was used as a loading control of the cytoplasmic proteins, and Histon 3 (His 3) was used as a loading control of nuclear proteins; (**d**). The relative values of EN-VCP in cytoplasmic and nuclear fractions, respectively; **(e**–**g**). The relative values of VCP-cofactors in cytoplasmic and nuclear fractions, respectively, including UFD-1 (**e**), NPL-4 (**f**), and P47 (**g**). The cytoplasmic proteins were normalized to the GAPDH, and the nuclear proteins were normalized to the Histon 3 (His 3). The relative value was presented by the fold of the WT young mice. *, *p* ≤ 0.05, **, *p* ≤ 0.01 vs. corresponding young mice; #, *p*≤ 0.05, ##, *p* ≤ 0.01 vs. age-matched WT mice.

**Figure 5 cells-10-02891-f005:**
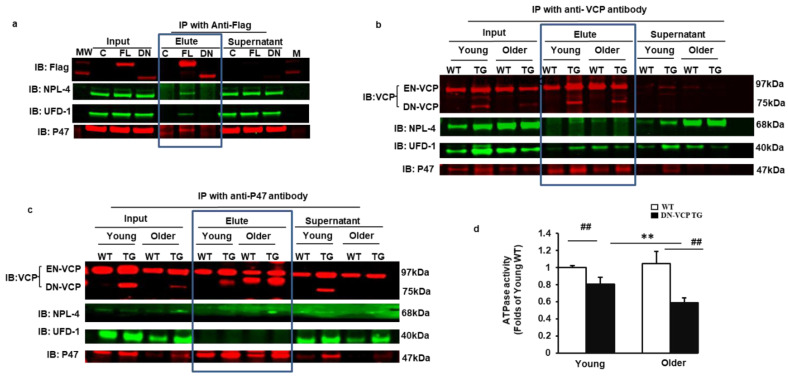
DN-VCP disrupts the VCP-cofactor interaction and leads to a binding preference of EN-VCP with the repressive protein P47. (**a**). Immonuoparticipation (IP) with anti-Flag and the Western blotting with the antibodies against the corresponding proteins in the cells overexpressed full-length VCP (FL-VCP) and dominant-negative VCP (DN-VCP) compared to the vector control. C: control plasmid with vector; FL: overexpressed full-length VCP, DN overexpressed DN-VCP; M: Molecular weight marker; (**b**,**c**). The representative images of the Western blots with the antibodies against the corresponding proteins after IP in cytoplasmic fractions of mouse heart tissues with anti-VCP (**b**) or anti-P47 (**c**). Input proteins were used as positive control and the supernatants of the IP were used as the negative control; (**d**). ATPase activity in DN-VCP TG and WT mouse heart tissues at young and older ages. **, *p* ≤ 0.01 vs. corresponding young mice, ##, *p* ≤ 0.01 vs age-matched WT mice.

**Figure 6 cells-10-02891-f006:**
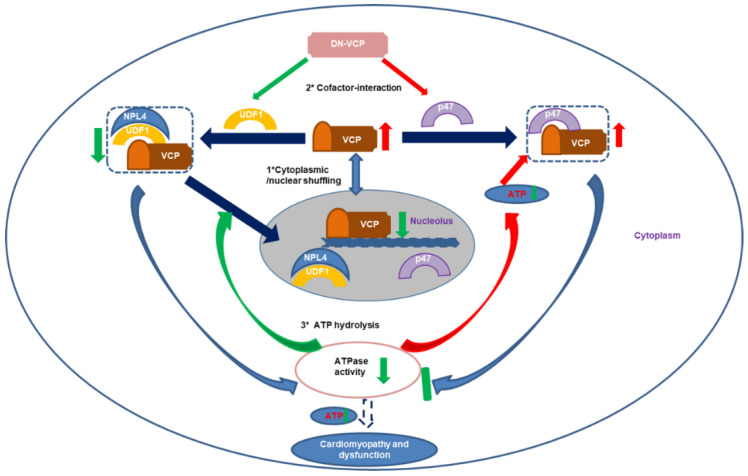
The summary of the findings: The results indicate that dominant-negative VCP (DN-VCP) acts as a dominant-negative mutation inhibiting VCP function in the mouse hearts through a comprehensive mechanism by impairing the endogenous VCP (EN VCP) and its co-factors cytoplasmic/nuclear shuffling, interfering EN-VCP-cofactor binding preference, and repressing ATP hydrolysis, resulting in a progressive dilated cardiomyopathy and dysfunction during the aging transition. Blue arrows: regular biological effects; Red arrows: increase or activating; Green arrows; decrease or inhibiting. UFD-1: ubiquitin fusion degradation protein 1; NPL-4: nuclear protein localization protein 4.

## Data Availability

The original images for western blots were provided in the main text.

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
