# Peer review of "Functional Inhibition of Valosin-Containing Protein Induces Cardiac Dilation and Dysfunction in a New Dominant-Negative Transgenic Mouse Model"

_cells, 2021, doi:10.3390/cells10112891_

Round 1

Reviewer 1 Report

Sun and colleagues conducted a very interesting study to investigate the in vivo role of Valosin-containing protein (VCP) in the heart. They generated a cardiac-specific transgenic mouse overexpressing a mutant of VCP bearing a deletion of N-domain of VCP to test whether overexpressed DN-VCP exhibits a functional inhibition of endogenous VCP. Moreover, they investigated the temporal effects of DN-VCP on cardiac morphology and contractile function in the transgenic model at baseline. Eventually they explored the molecular mechanisms underlying the adverse effects of DN-VCP observed in mouse hearts.

Their results showed new possible molecular pathways that could be involved in cardiomyopathy and cardiac dysfunction during the aging transition. So we can define this study an important proof of concept of the VCP role in the cardiac environment and in cardiomyocytes.

It would be very useful investigate the effect of physical stress in this transgenic model to deepen the physiological role of VCP in the cardiac function.

Overall the authors presented exhaustively their results, and only few improvements are needed.

In the introduction session the author anticipated their results (lines 92-96). This part should be move to the results or the conclusions sessions.

Furthermore, they should add more details about their methods and the protocols applied during the study.

They should spell out the abbreviations used in the Echocardiography paragraph, and better define the abbreviation about DN-VCP: N-domain VCP and Dominant Negative VCP can be confused.

Reviewer 2 Report

This manuscript by Sun et al described a new VCP dominant-negative transgenic (DN VCP TG) mouse model, which showed a functional inhibition of  VCP in the heart tissue without changing the expression of endogenous VCP (EN-VCP) level. Inhibition of VCP activity resulted in cardiac dilation and progressively decline in heart function.  The Authors also explored the potential molecular mechanisms involving the inhibition of DN-VCP in EN-VCP activity, the inferences in subcellular translocation and cofactor-interaction, and age-related adaption.

The phenotype of this mouse model is interesting and significant since the VCP KO is lethal; thus, the DN-VCP TG mouse provides a new alternative strategy to investigate the role of VCP in cardiac function in both physiological and pathological the heart. The timing of the functional deficit correlates with the changes in expression and nuclear localization of EN-VCP and its binding partners. Overall, this is a very well-conducted study. The experiments are appropriate, and the data logically support the conclusions. A few minor comments may need to be  addressed:

  1. To better understand the dominant-negative effect of the DN VCP, the authors should provide more information on the domains of VCP, either expand the description or make a diagram, such as the main function of the N- domain, especially about the binding partners chosen for study here, how many amino acids have been terminated, and whether such an N-terminal domain would be expected to affect the catalytic activity of the mutated form of the ATPase.
  2. Fig 6 can be improved to clarify further the subsequence of the multiple processes in regulating the progressive cardiac dysfunction during aging. The corresponding explanation may be needed in the conclusion sections.
  3. All the abbreviations in the figures need to be fully termed in the figure legends.
